# Experiences of Barriers to Self-Compassion in Women Experiencing Weight Difficulties: An Interpretative Phenomenological Analysis

**DOI:** 10.3390/jpm12091504

**Published:** 2022-09-14

**Authors:** Anna Jeziorek, Afsane Riazi

**Affiliations:** 1Department of Psychology, Royal Holloway, University of London, Egham TW20 0EX, UK; 2Department of Psychology, Richmond American University in London, London W4 5AN, UK

**Keywords:** self-compassion, women, overweight, weight loss, weight management

## Abstract

There is currently little understanding of why some individuals find it difficult to engage in self-compassion (SC), defined as a form of self-kindness, self-acceptance, and courage to face one’s distress. This is especially true for women experiencing weight difficulties, despite the emergence of promising results from compassion-focused approaches to weight management. Consequently, this study undertakes a qualitative study to explore the experiences of barriers to self-compassion in women who were actively trying to manage their weight, using interpretative phenomenological analysis (IPA). A qualitative study was employed using an interpretive approach. Using purposive sampling, 10 women were recruited from London-based weight loss groups. Three super-ordinate themes emerged: (I) feeling unable to prioritise own needs over others’ needs; (II) having to learn and sustain a new approach to weight loss; and (III) having very high standards. The emergent themes suggest that women who face weight difficulties have numerous barriers to self-compassion. To fully utilise compassionate-based weight loss interventions for women, it is important to recognise these barriers and implement strategies to lessen their impact.

## 1. Introduction

The trend of increased obesity is becoming worse due to complex biological, psycho-social, and environmental factors [1]. Traditional approaches to weight loss, such as restrictive eating and exercise regimes, are not sustainable, as they often cause rapid weight re-gain [2]). There is an increasing focus on person-centred diets, lifestyles, and other solutions to losing weight [3]. Targeting self-compassion (SC) is thought to be one of the innovative ways that could assist with problematic eating patterns and other factors related to being overweight.

Derived from evolutionary science, attachment theory, and cognitive neuroscience, self-compassion (SC) is a form of self-kindness, self-acceptance, and courage to face one’s distress [3]. It is a way of relating to the self by: (1) treating oneself with self-kindness; (2) acknowledging that suffering is a part of common humanity; and (3) mindfulness, by accepting suffering and holding it in awareness [4]. SC mediates the relationships between Body Mass Index (BMI) and the symptoms of pathological eating [5], and links positive body image [6], increased body compassion [7] and improved exercise motivation [8]. SC is associated with less negative attitudes towards overeating following an unsuccessful diet [9], less weight stigma [10], and decreased binge eating [11]. Based on the protective role of SC, innovative compassion-focused interventions have been developed for weight difficulties, e.g., BEFree, a group-based compassion-based weight management programme with elements of psycho-education, mindfulness, and compassion [12]. Compared to wait-list controls, BEFree reduced overeating patterns as well as shame, self-criticism, and negative internal states. Another compassionate-based group intervention [13] significantly improved quality of life, physical exercise, and SC skills and also reduced problematic eating patterns, BMI, and weight-related self-stigma and self-criticism in women with weight difficulties [13,14]. Despite the encouraging findings, some participants did not respond to the SC techniques or found them difficult to implement (e.g., [14]).

In other populations, people experiencing depression reported perceiving SC as meaningful and useful, though difficult and unfamiliar to implement [15]. In both clinical (depression) and non-clinical samples, a strong association between fears of SC for self and self-criticism, insecure attachment, depression, anxiety, and stress have been suggested [16,17]. Factors that may stand in the way of cultivating SC include the perception of receiving it as a sign of weakness; painful memories of not receiving it when needed; feeling unworthy of it [18]; and feeling overwhelmed by sadness and grief when experiencing warmth and kindness [19].

The present study attempts to explore the above barriers further in a non-clinical sample of overweight women actively trying to lose weight. Previous studies have focused on barriers to SC in a population experiencing eating disorders but not in overweight individuals (e.g., [14,20]. The exception has been a qualitative study that suggested that overweight individuals in clinical settings, who broke diet regimes, faced high levels of self-hate and self-disgust, which impacted their ability to be self-compassionate [3]. The present study attempts to explore the above barriers further in a non-clinical sample of women with weight difficulties recruited from the community. Moreover, research on SC and eating regulation has typically used quantitative methods (e.g., [21]. The experiences of the process of engaging in SC, and overcoming any obstacles standing in the way of cultivating it, have not yet been qualitatively explored in overweight women in the community. This qualitative study explores the experiences of SC in women who were overweight and, following a number of previous weight loss attempts, were actively trying to lose weight, using interpretative phenomenological analysis (IPA). The study will aim to address the following research questions:(1)How do overweight women think about the experience of self-compassion?(2)How do women with problematic eating perceive difficulties in experiencing self-compassion?

## 2. Materials and Methods

### 2.1. Participants

The participants were recruited using purposive sampling, consistent with the qualitative paradigm and IPA approach [12,22]. Overall, 10 women met the study criteria and agreed to be interviewed. The women’s ages ranged from 21 to 70, with the majority of women (70%) being aged between 29 and 39. A total of 60% of women described their descent as “White British”, and the remaining 40% responded as either “Black British” or “Others”. Further, 60% of the participants had at least an undergraduate degree, and the remaining 40% reported having an education at A levels. The majority of respondents reported being in employment (70%), and the remaining 30% were either unemployed or in full-time education. No participants dropped out of the study.

### 2.2. Sampling and Recruitment

All participants were over 18 years old, female and fluent in English, had a Body Mass Index (BMI) over 25 kg/m^2^ (BMI > 25), and managed their excess weight using a community-based intervention. The BMI criteria were selected based on the UK data, suggesting that almost 70% of the adult population is reported to have a BMI over 25 (overweight or obese; [23]. All women had a history of at least three dietary attempts. They were recruited through local weight loss groups. Monetary incentives were introduced in the form of £10 vouchers. Both authors were females. At the time of the recruitment, the first author was undertaking doctoral training in clinical psychology and was employed as a trainee clinical psychologist at various nationally funded (NHS) clinics in London. The second author was a senior lecturer in psychology at the Department of Psychology with extensive expertise in qualitative research methods and previous publications on issues of obesity.

### 2.3. Procedure

The core components of a semi-structured interview were developed by drawing on the existing compassion research (e.g., [2,3]. These were used to form specific questions which were discussed with two weight loss group facilitators and two volunteers. Their feedback informed the final set of questions in terms of relevance and ability to elicit the experiences of participants. The study was approved by the Royal Holloway Research Ethics Committee (Project ID: 868). The participants provided written informed consent to the inclusion of fully anonymised material related to themselves, and they acknowledged that they could not be identified via the article. The interviews lasted between 35 and 95 min, were audio recorded, and transcribed verbatim.

### 2.4. Methods

An interpretative phenomenological analysis (IPA) approach was used in this study. IPA is a qualitative approach involving a comprehensive examination of the details of lived experiences. More precisely, it is an in-depth interpretation of each participant’s experience and the meaning that the experience holds for the individual [24]. IPA is particularly useful for understanding under-researched phenomena or perspectives and suitable for identifying and understanding meaningful subjective experiences or perspectives [25].

### 2.5. Data Analysis

Data were collected from ten in-depth semi-structured interviews and analysed using IPA [25]. Firstly, the researcher immersed herself in the data by reading and re-reading the original transcripts, ensuring an active phase of engagement with the data. This stage focused on descriptive comments (the content of the participant’s answer), linguistic comments (e.g., pauses, laughter, etc.), and conceptual comments (focusing on a more abstract level and using tentative interpretations). The three types of comments included summarising, making connections between what was said, noticing contradictions in what the participant said, and interpreting what it could have meant for the interviewee. Following this, the researcher made notes of anything that was of interest within the text, exploring participants’ language and the context of their experiences, as well as identifying more abstract concepts around the patterns of meaning. The next step focused on developing a large number of emerging themes and was more interpretative in nature, as it reflected the participant’s original phrases and also the researcher’s interpretation. The fourth step involved generating connections between the emerging themes, grouping similar ones together, and discarding irrelevant ones. Following the grouping stage, the emerging themes were fitted together to create a structure that would point to the most interesting and important aspects of the participant’s account ([25], “p.96”). Themes were then abstracted and synthesised to assist with clustering super-ordinate themes for each participant.

### 2.6. Quality and Validity

Four general principles were consulted to guide the current study: sensitivity to context, commitment and rigour, transparency and coherence, and impact and importance [26]. Sensitivity to context was demonstrated in various stages of the study; a purposive sample of participants who shared a particular lived experience was employed, and the researcher constantly used her empathic engagement skills and was highly attuned to what was being said, helping the researcher to probe and discover new and important aspects [24]. To ensure the commitment and rigour of an IPA study, a measure of prevalence for a theme was provided. To maintain coherence and transparency, the researcher: provided clear descriptions of all study stages, arranged frequent consultations with supervisors, looked jointly at transcripts and emerging themes, used extracts from transcripts to support themes, and used a reflective journal to track the perspectives and contributions to the process.

## 3. Results

The data analysis revealed ten subordinate themes (see Table 1), which were grouped into three super-ordinate themes. The nature of each super-ordinate theme was summarised by a number of subthemes, and each subtheme was supported by a selection of the most representative quotes [25].

(1)“I don’t think I really consider myself enough”

The first super-ordinate theme revealed that, whilst attempting to lose weight, the majority prioritised other responsibilities over being self-compassionate. The reasons for this were multi-fold. All participants reported that being self-compassionate meant losing weight by eating healthily and having an exercise routine. However, other responsibilities stood in the way of achieving this. Important in the participants’ lived experiences were commitments, such as holding a job, working irregular hours when studying for a degree, or looking after the house.
“I think my biggest problem is just life getting in the way and I just prioritise everything over that”(Participant 9)

Life could get in the way of not just losing weight but also maintaining new weight:
“… you may lose a stone or a stone and a half […] and you feel good but then just life, you put on [weight], and children, grabbing things to eat on the way, yeah, I think life generally”(Participant 7)

The tendency to prioritise daily commitments over a self-compassionate attitude to weight loss coincided with prioritising other people’s needs. The majority felt pressured to take care of their loved ones, meaning there was no time to rest, focus on their own nutritional needs, or attend exercise classes. Some felt unable to afford time or finances to meet their health needs:
“I don’t think I really consider myself enough […]; I put myself at the bottom of the list every time”(Participant 3)

The motivation to prioritise others’ needs was sometimes difficult to explain;
“I’m just like a robot and I just […] do things but, I suppose, people do need to do things for themselves and I don’t do enough at all for myself, in any way”(Participants 3)

Women spoke about a sense of guilt they felt when attempting to focus on their weight loss in a compassionate way. This was represented by negative thoughts stating they should be doing something else than looking after themselves:
“… not feeling guilty about that after you’ve done that so, kind of, giving yourself permission to do it”(Participant 4)

(2)Re-learning a new way of life

This super-ordinate theme comprised three subordinate themes, and each subtheme related to a different aspect of difficulties with cultivating a self-compassionate approach to weight loss, ranging from the novelty of experiences to the participants’ perspectives on their past and reflections on less accessible (unconscious) material. Many had practised being more self-compassionate at some point in their lives. However, facilitating SC whilst trying to lose weight involved learning a new skill and took time and practice.
“… it’s a very new thing, it’s odd, that what I say to myself […] it’s learning a new way of life, it’s interesting […]it is odd because it’s out of my comfort zone and I’ve lived a certain way for so long, now I’m changing it”(Participant 8)

Being self-compassionate meant that compared to restrictive dieting, the process of slimming down could take time and be gradual and involve more self-awareness;
“… back then [when crash dieting] it was very much all or nothing, either you’re hitting the numbers or you’re not, but equally then you’re not being able to be very self-aware, not noticing the progress, […] whereas now being more compassionate, just being a bit more self-aware”(Participant 4)

Despite difficulties cultivating more SC, some reflected on occasions when they were able to change their attitudes to dieting, which made them kinder to themselves;
“I’m definitely seeing things in a different light, like I’m not just doing it for a weight loss or whatever. It’s more about feeling comfortable with myself”(Participant 9)

All participants reflected on how their early experiences may have impacted their current ability to be self-compassionate in relation to weight management. Some described experiences of emotional and physical abuse or bullying at school.
“I had quite a traumatic childhood and I think that may have been the part of me, developing it [being overweight], because of the stress, and the stuff I underwent, I think. And I think this was also what made me comfort eat, maybe, and becoming this shape”(Participant 1)
“…sometimes you get a few comments when you’re at school, when you’re a bit younger people are a bit nasty”(Participant 5)

For some, the difficulty with practising SC stemmed from a lack of role modelling in the family home;
“I think it was probably hard to model or to be self-compassionate without really having that model. I think people were quick to say when something was wrong and to call it out and criticise than saying when something was going right […]”(Participant 4)

Finally, some had not previously thought about SC and what may have contributed to not practising it. Discussing SC was novel and impacted their ability to reflect on it;
“I just can’t work out what the block is, which is really difficult because it’s not something I’ve ever thought about”(Participant 3)

(3)“I have very high standards for myself”

The third super-ordinate theme revealed the impact that high standards had on the participants’ lives and the way they managed their weight, which originated internally and externally. Unrelenting standards dominated how some women perceived themselves and maintained their desires to have perfect eating patterns to help them become slim.
“…my desire was just to be slim and to be this person that I had envisaged in my head, that I’d never been before. […] It’s an obsession. […] it’s just a sheer drive to achieve this perfection but you’ve no idea what this perfection is”(Participant 8)

Participants with perfectionist traits felt that they needed to earn SC:
“If I see that […] I’ve lost weight, I look great, […] I will maybe be a bit more nice to myself, but not now, not yet”(Participant 10)

The high standards for themselves also contributed to the judgement received from themselves and others. The internal critical voice commented on their appearance and weight and became even stronger if they failed at weight loss.
“…I’ve definitely yo-yoed a lot in the last 10 years, definitely. […] If I feel like my clothes are getting a bit tighter, […] I could have put 4, 5, 6 pounds and I’m like ‘oh my God, this is awful, what are you doing, you’re so fat, come on!’(Participant 5)

The lived experiences of feeling judged related to the comments received from their families, friends, or weight-loss group facilitators.
“… people comment on it [weight] a lot so it further makes me uncomfortable going out, going to occasions such as weddings and things like that because I know there will be comments about it”(Participant 2)

Part of the participants’ unrelenting standards was reflected in the way they compared themselves unfavourably with others:
“I was always aware that I was bigger than my friends because a lot of my friends are […] so petite, so so slim”(Participant 6)

Some spoke about appearance and weight comparisons with people on social media or in the press:
“I think that’s why you end up torturing yourself a little bit because you always aspire to being like that, to fit into that category, or the images they’re portraying of people in the media […]”(Participant 7)

Finally, some reflected that if they were to facilitate more SC whilst attempting weight loss, they were too lazy or soft. Many had concerns that experiencing more SC meant they were not motivated enough to lose weight, or were selfish.
“… in some situations, if I gave myself any more SC, I would just lie down. I’m not sure if it would mean that I’m going to be too soft”(Participant 3)

## 4. Discussion

The current study explored the experiences of barriers to SC in overweight women. The analysis revealed three interlinked super-ordinate and ten subordinate themes (Table 1). Central to the experiences of barriers to SC was that the participants understood that cultivating SC had an important positive effect on their weight management, e.g., feeling optimistic about losing weight in a more sustainable way and stopping the vicious circle of restricting their food intake and overeating. SC helped to focus less on food-related thoughts, better plan their meals and reduce self-criticism and other negative effects related to being overweight. However, one of the key findings, suggested in the first super-ordinate theme, was the conflict between practising SC and fulfilling other responsibilities. Commitments, such as work, studies and/or being responsible for others, led to reduced self-care activities and possible weight gain. They also reported guilt when practising self-care activities, reflecting the challenges women still face in Western society for meeting others’ needs and feeling unable to negotiate others’ demands [27].

The second super-ordinate theme suggested women found that developing a more self-compassionate stance towards themselves required time and effort. Although many women perceived it as useful, they found it difficult to implement, which mirrored other research on barriers to SC [2]. Women also described oscillating between habitual hostile self-judgements and new, more self-accepting thoughts, which is in line with the previous qualitative research on self-hatred and SC (e.g., [3,16,28].

The majority reported some emotional neglect and weight-related bullying in childhood and reflected that these traumatic experiences may have impacted their ability to be self-compassionate. Survivors of physical and emotional maltreatment and emotional neglect report lower levels of SC compared to those with no abuse history [29]. The current study supports the view that women who experienced emotionally adverse events may have difficulties cultivating SC and maintaining positive self-beliefs [30] due to a fear of SC [31], SC bringing back sadness and grief, or feeling unworthy of it [19], which has been suggested in previous studies on barriers to SC.

Some potential unconscious mechanisms may be at play when engaging with SC. A few women were unable to access their cognitions when discussing barriers to SC. Individuals who struggle with weight-related shame and self-criticism may have limited access to their internal experiences (Kolts, 2016). Participants may need more time to reflect on and develop language and skills in their self-compassionate practice. Future research may help to further investigate compassion-related unconscious mechanisms.

The third and final super-ordinate theme suggested that women had very high standards for themselves regarding their weight and self-image. Some strove for perfection when starting a new food or exercise plan and felt any lapse equalled failure. This rigid, ‘all-or-nothing’ approach prevented self-kindness and the cultivation of self-care. The inverse effect between maladaptive perfectionism and SC has been documented before [32,33].

The majority of women developed restrictive eating patterns and exercise regimes, such as ‘crash diets’ related to extrinsic goals (e.g., attending a wedding). These were not sustainable and eventually led to overeating and a sense of failure. Some went through multiple failed attempts before realising they were stuck in a vicious circle of intentional rapid weight loss and weight gain [2]. This vicious cycle may be related to the negative pathway of eating regulation where extrinsic weight-related goals are positively associated with ‘fat talk’ (e.g., body concerns and body comparisons) and negatively associated with SC, which may lead to unhealthy eating behaviours [34]. In addition, individuals who pursue appearance goals may be more likely to have binge eating episodes [35].

Women’s lived experiences of SC included frequent judgement from self and others. As seen in [3], some interviewees were hostile toward their self-image when confronted with their weight, struggled to lose weight or gained weight. Previous research suggested that people who try to lose weight may be particularly prone to weight self-stigma due to frequent feelings of failure and self-criticism [3,9].

Although the negative impact of comparing their weight and appearance with friends, family, or on social media was acknowledged, the women in the current study seemed unable to cease this behaviour, possibly due to low self-esteem. A strong positive link has been found between low esteem and being overweight [36]. People with low self-esteem engage in downward social comparisons to increase their self-worth [37]. The motivation to protect a fragile self-image can lead to a rigid and negative mindset, unable to tolerate alternative viewpoints [38].

### 4.1. Limitations

The study had a few limitations. Although the sample was relatively homogenous, the age range was wide (18–74), and the older participants lacked familiarity with the concept of SC. However, this was managed by explaining SC in jargon-free terms, such as ‘self-kindness’, ‘courage’, and ‘looking after self/health’. Furthermore, there was no cut-off point for the number of dietary attempts; some dieted on and off for years. Due to a lack of findings on the differences between females who attempted to lose weight a few times as opposed to those who dieted on and off throughout their life, it was difficult to conclude whether this factor impacted the results.

### 4.2. Clinical Implications

The high overweight rates in the UK and the lack of evidence for existing approaches to weight loss [2] point to finding new ways of helping people sustainably lose excess weight. Further developments in innovative interventions are crucial [39]. Compassion-based weight loss programmes can be of clinical importance if their effectiveness is well explored and improved, and this study provides a step towards personalised interventions. For example, new community-based interventions for women could include psycho-education about the importance of prioritising one’s health needs if women want to achieve sustainable weight loss. Additionally, helping women to view a self-compassionate mindset as a lifestyle change (rather than another diet) and tackling maladaptive weight-related shame, rigid thinking, and perfectionism could be tailored to women’s individual needs. The improved understanding of SC could be extended to help women tolerate and deal with weight-related distress and stigma [40]. A self-compassionate stance, including emotional support around self-judgement, could be promoted in commonly used weight management practices, such as GPs and local weight loss groups, which currently give out advice to restrain food intake and increase exercise levels. Furthermore, future research could build on the current qualitative study using other methodologies, including mixed and quantitative (e.g., cross-sectional), to further elaborate on the validity and reliability of the constructs identified. Future findings on understanding SC and its positive impact on weight management could influence public policies and campaigns related to sustainable weight loss in women.

## 5. Conclusions

The current study offered new insights into the experiences and meaning of SC in overweight women who, following previous attempts at slimming down, were actively trying to lose weight using community-based approaches. This study suggested a number of internal and external barriers to SC, including prioritising other needs over self-care, difficulties in practising a self-compassionate stance in day-to-day life, and having unrelenting standards for themselves. Further investigations are needed to better define, validate, and understand these constructs. The findings could inform policy and practice in designing and delivering well-informed compassionate-based weight management programmes for overweight women.

## Figures and Tables

**Table 1 jpm-12-01504-t001:** Master table of themes.

Super-Ordinate Theme	Subordinate Theme
“I don’t think I really consider myself enough”	Prioritising other needs over SC
	Prioritising other people over SC
	Feeling guilty of prioritising SC
Re-learning a new way of life	“It’s a very new thing, it’s odd”
	The impact of growing up
	Unconscious barriers to SC
“I have very high standards for myself”	Striving for perfection
	Judgement from self and from others
	Comparing yourself to others
	Negative perceptions of SC

## Data Availability

The data that support the findings of this study are available on request from the corresponding author, [A.J.]. The data are not publicly available due to restrictions (data contain information that could compromise the privacy of research participants).

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
