# Peer review of "Experiences of Barriers to Self-Compassion in Women Experiencing Weight Difficulties: An Interpretative Phenomenological Analysis"

_jpm, 2022, doi:10.3390/jpm12091504_

Round 1
Reviewer 1 Report
Dear authors:
the paper is of a high quality one. The topic is very interesting and inspiring. I would like to highlight the quality of Introduction, Methods used in presented paper and references.
Authors are recommended to expand conclusion by implementing more outcomes of presented research.
Author Response
Reviewer 1: Authors are recommended to expand conclusion by implementing more outcomes of presented research.
We have expanded the conclusion by expanding the following paragraph:
Compassion-based weight loss programmes can be of clinical importance if their effectiveness is well explored and improved and this study provides a step towards personalized interventions. For example, new community-based interventions for women could include psycho-education about the importance of prioritising one’s health needs if women want to achieve sustainable weight loss. Also helping women to view self-compassionate mind-set as a lifestyle change (rather than another diet) and tackle maladaptive weight-related shame, rigid thinking, and perfectionism could be tailored to women’s individual needs. The improved understanding of SC could be extended to help women tolerate and deal with weight-related distress and stigma (Mantzios & Egan, 2017). A self-compassionate stance, including emotional support around self-judgement, could be promoted in commonly used weight management practices such as GPs and local weigh loss groups which currently give out advice to restrain food intake and increase exercise. Furthermore, future research could build on the current qualitative study using other methodologies, including mixed and quantitative (e.g. cross-sectional) to further elaborate the validity and reliability of the constructs identified. Future findings on understanding of SC and its positive impact on weight management could influence public policies and campaigns related to sustainable weight loss in women.
Reviewer 2 Report
The study is unacceptable, as it only covers 10 cases. It is unlikely to draw far-reaching conclusions based on interviewing 10 people! Important statistical comparisons and inference are missing. Moreover, where are the numerical results for the aforementioned population? It seems that the authors have relied on a qualitative study, but it doesn't really contribute anything, and it's not clear why such descriptions were created. Comparisons are missing, for example, in terms of BMI.... The authors describe them in the methodology, and then it is not used for anything.
After the discussion section, the section on the limitations of the survey is missing, and these are many, from the sample size to the lack of application of statistical corrections and the failure to present numerical results in the results section.
In addition, please note whether the bibliographic references in the text comply with MDPI requirements.
Author Response
Reviewer 2:
This was a qualitative study and therefore no statistical comparisons or inferences were performed and no numerical results were obtained. As this was a qualitative study, it is impossible to generalize the results to the general population. Also, a sample of 10 was in the top range of what is suggested (5-6) for this type of methodology (Smith et al., 2009).
Reviewer 3 Report
This phenomenological study focused on the experiences of self compassion in overweight females who followed a number of previous weight loss attempts, were actively trying to lose weight. They used Interpretative Phenomenological Analysis (IPA) to explore related themes and subthemes on 10 subjects. Their thoughts and perception of self-compassion experience during the process have been searched for. They found three super-themes from these interviews. Interpretative phenomenological analysis have been chosen and used appropriately in the study. The sample selection and methods used have been carefully described and discussion has been made accordingly. In general study is a well designed and carefully written study which would be useful for researchers and readers in this issue. I suggest minor revisions;
1. Please add if there is any drop-out in the study.
2. Were the transcripts of the interview returned to the participants for comment or correction?
3. Define the characteristics of the researcher(s) , gender, occupation, experience, training.
Author Response
Reviewer 3:
- Please add if there is any drop-out in the study.
Response. No participant dropped out of the study and this information has now been added to the methodology section.
- Were the transcripts of the interview returned to the participants for comment or correction?
Response. Although it is encouraged in qualitative studies that the transcripts of the interviews are returned to the participants for any comments or correction, this was not possible on this occasion due to time restrictions in order to meet deadlines for a completion of the first author’s doctoral programme.
- Define the characteristics of the researcher(s) , gender, occupation, experience, training.
Response: Both authors were females. At the time of the recruitment, the first author was undertaking a doctoral training in clinical psychology and was employed as a Trainee Clinical Psychologist at various nationally funded (NHS) clinics in London. The second author was a Senior Lecturer in Psychology at the Department of Psychology with an extensive expertise in qualitative methods and previous publications on issues of obesity. This information has now been added to the methodology section.
Round 2
Reviewer 2 Report
I still have big doubts with methodology of the paper. I encourage Authors to present the resuls numerically at least with one table.